# IMPROVED TRAINING TECHNIQUES FOR ONLINE NEURAL MACHINE TRANSLATION

## ABSTRACT

Neural sequence-to-sequence models are at the basis of state-of-the-art solutions for sequential prediction problems such as machine translation and speech recognition. The models typically assume that the entire input is available when starting target generation. In some applications, however, it is desirable to start the decoding process before the entire input is available, *e.g.* to reduce the latency in automatic speech recognition. We consider state-of-the-art "wait-$k$" decoders, that first read $k$ tokens from the source and then alternate between reading tokens from the input and writing to the output. We investigate the sensitivity of such models to the value of $k$ that is used during training and when deploying the model, and the effect of updating the hidden states in transformer models as new source tokens are read. We experiment with German-English translation on the IWSLT14 dataset and the larger WMT15 dataset. Our results significantly improve over earlier state-of-the-art results for German-English translation on the WMT15 dataset across different latency levels.

## 1 INTRODUCTION

Sequence-to-Sequence (S2S) models are state-of-the-art for tasks where source and target sequences have different lengths, including automatic speech recognition, machine translation, speech translation, text-to-speech synthesis, *etc*. The most common models are composed of an encoder that reads the entire input sequence, while a decoder (often equipped with an attention mechanism) iteratively produces the next output token given the input and the partial output decoded so far. While these models perform very well in the typical *offline* decoding use case, few studies consider how S2S models are affected by low-latency constraints, and which architectures and strategies are the most efficient. Low-latency decoding is desirable for applications such as online speech recognition, and as-you-type machine translation. In such scenarios, the decoding process starts before the entire input sequence is available, and the output sequence is produced in an on-the-fly manner. However, if we consider for instance machine translation, online prediction generally comes at the cost of reduced translation quality and more research is needed to reach the grail of natural and high-quality online speech-to-speech interpretation.

In this paper we consider deterministic "wait-$k$" decoders that are state of the art for low-latency decoding (Ma et al., 2019; Zheng et al., 2019b). These decoders first read $k$ tokens from the source, after which they proceed to alternatingly produce a target symbol and read another source symbol. We compare two architectures to implement such models: one based on a 2D-convolutional sequence-to-sequence model (Elbayad et al., 2018), and one based on the attention-based transformer architecture (Vaswani et al., 2017). For these models, we investigate the impact of the choice of $k$ when training the models, and when using them to generate translations. For the transformer model, we also consider the effect of updating the hidden states of previous target symbols based on the full source context that is available at any moment. These updates are inspired from the 2D convolutional model, where such "updates" are an inherent consequence of the architecture.

In summary, our contributions are the following:

1. We compare transformer and 2D convolutional architectures for online machine translation.
2. We propose improved training techniques for wait-$k$ by first using uni-directional encoders and training across multiple values of $k$

3. We boost the translation quality in test-time by updating the prefix hidden-states in transformer decoders.
4. We set a new state of the art for online translation on WMT15 German-English, improving over the recent state-of-the-art results by Ma et al. (2019) and Zheng et al. (2019b) across a full range of latency levels.

The rest of this paper is organized as follows: Section 2 presents related work on low-latency machine translation; Section 3 details our low-latency models, which we evaluate in Section 4 for German-English machine translation on the IWSLT14 dataset as well as the larger WMT15 dataset. Section 5 concludes the paper.

## 2 RELATED WORK

In order to study differences between translated and simultaneously interpreted text, He et al. (2016a) produced a parallel corpus between both and showed that human interpreters regularly apply several tactics to reduce translation latency, including sentence segmentation and passivization. In machine translation, after pioneering work from Fügen et al. (2007), Yarmohammadi et al. (2013) and He et al. (2015) proposed methods to increase the alignment monotonicity for statistical machine translation. To allow for a more flexible segmentation, Grissom et al. (2014) introduced a trainable segmentation module that decides whether to write or read tokens based on the prediction of the final verb and the next token in the source sequence. Similarly, Oda et al. (2015) use output of a syntax-based statistical translation system to find the optimal segmentation strategy that maximizes the quality of the translations via greedy search and dynamic programming.

One of the first works on online translation to use attention-based sequence-to-sequence models is that of Cho & Esipova (2016), which uses manually designed non-trainable waiting criteria that dictate whether the model should commit to a read/write operation. The neural transducer of Jaitly et al. (2016) reads equally-sized chunks of the source sequence and generates output sub-sequences of variable lengths, each ending with a special token marking the end of the writing. For training, a single segmentation is chosen to optimize the likelihood of the full output sequence. Raffel et al. (2017) propose an alternative to attention using monotonic alignments. Whereas attention requires access to the full source sequence to compute the weights, monotonic alignments enable linear time computation of the weights and online decoding.

Another line of research treats general alignments of source and target sequence as a latent variable. Inspired from the HMM word alignment model used in statistical machine translation (Vogel et al., 1996), the segment-to-segment neural transducer model of Yu et al. (2016) integrates a latent alignment variable into an LSTM-based sequence-to-sequence model where the transition probabilities are conditioned on the encoder-decoder hidden states. During training the alignment is marginalized out with a forward-backward algorithm (Rabiner, 1989). However, their approach is used as an alternative to attention-based models, not for low-latency translation. Luo et al. (2017) introduce a recurrent sequence-to-sequence model with binary stochastic decision variables to either emit the next output token or advance in encoding the source sequence with a unidirectional LSTM that jointly encodes the partially generated output sequence. The stochastic decisions are optimized using a standard policy gradient reinforcement learning approach. Gu et al. (2017) propose a trainable agent emitting read/write decisions modeled as a recurrent neural network fed with the encoder and decoder current hidden states. In their framework, a left-to-right recurrent sequence-to-sequence model is first pre-trained on the full bi-texts and then fine-tuned with policy gradient to optimize a reward balancing the quality and the latency of the translations.

Dalvi et al. (2018) used a static decoding algorithm that starts with $k$ read operations then alternates between blocks of $l$ write operations and $l$ read operations. Albeit simple, this approach outperforms the information based criteria of Cho & Esipova (2016) and allows for complete control of the translation delay. Their attempt to integrate incrementality in the training by pre-aligning the source-target sequences and then attending over a constrained span of source positions failed to improve the translation quality. The work of Press & Smith (2018), although not tackling the simultaneous translation task, allows for emitting target tokens before reading the full input sequence. Similar to the incremental training of Dalvi et al. (2018), it requires pre-aligning the bitexts. More recently, Ma et al. (2019) trained a sequence-to-sequence model based on the transformer architecture (Vaswani et al., 2017) with an integrated "wait-$k$" agent that first reads $k$ source tokens then alternate single

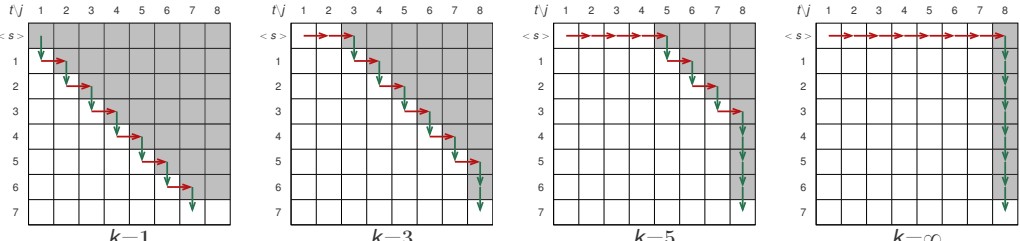

Figure 1: Low-latency decoding as a sequence of reads (green) and writes (red) over a grid spanned by the source (horizontal) and target sequence (vertical). Different panels show decoding paths defined by the nr. of initial reads $k$, of the wait-$k$ decoder. For training we optimize the emissions probabilities for all writes on a single or multiple paths, or for all shaded positions above the path.

read-writes, similar to Dalvi et al. (2018) but using $l = 1$. Wait-$k$ approaches, were found most effective by Zheng et al. (2019b) when trained for the specific $k$ that is used to generate translations. This, however, requires training separate models for each potential value of $k$ used for translation. As an alternative, Zheng et al. (2019a;b) both learn an adaptive policy to produce read/write decisions in a pre-trained offline translation model. Zheng et al. (2019a) train the model for decoding along two wait-$k$ paths and Zheng et al. (2019b) use supervised training based on an oracle read/write sequence derived from the pre-trained offline translation model. Different latency levels are achieved by thresholding the policy's confidence, however, on most regimes their model performs worse than using transformer models directly trained for wait-$k$ online translation. In our work, we also use wait-$k$ decoders, but unlike Ma et al. (2019); Zheng et al. (2019a;b) we opted for uni-directional encoders and show that they are efficient to train in an online setup and can achieve better results if we update the prefix decoder states during inference.

## 3    LOW-LATENCY SEQUENCE-TO-SEQUENCE MODELING

Let $(\boldsymbol{x}, \boldsymbol{y})$ be a pair of source-target sequences of respective lengths $|\boldsymbol{x}|$ and $|\boldsymbol{y}|$. Low-latency decoding consists of executing a sequence of interleaved reading and writing operations, consuming tokens from the source $\boldsymbol{x}$ and producing tokens of the target $\boldsymbol{y}$. Low-latency decoding paths can be represented on a $|\boldsymbol{y}| \times |\boldsymbol{x}|$ grid, where we advance from the top left to the bottom right in a total of $|\boldsymbol{x}|$ read steps and $|\boldsymbol{y}|$ write steps. See Figure 1 for an illustration.

We consider two recent sequence-to-sequence models for low-latency decoding: the transformer architecture of Vaswani et al. (2017) and the 2D convolutional architecture of Elbayad et al. (2018). We describe in Section 3.1 how each of these architectures is adapted for the task of simultaneous translation. In Section 3.2 we describe how these architectures can be trained for online decoding, based on one or more decoding paths.

### 3.1    LOW-LATENCY DECODING ARCHITECTURES

To formalize the low-latency decoding process we model it as a sequence of $|\boldsymbol{y}|$ write steps. At step $t \in \{1, \ldots, |\boldsymbol{y}|\}$ we decode the $t$-th target token $y_t$, conditioning on the target prefix $\boldsymbol{y}_{<t}$ and the source sequence read up that point $\boldsymbol{x}_{\leqslant z_t}$. The value of $z_t$ is dictated by the decoding path.

The decoder computes the distribution over the next target token given the source and target contexts,

$$p_\theta(y_t | \boldsymbol{y}_{<t}, \boldsymbol{x}_{\leqslant z_t}, \boldsymbol{z}_{<t}), \tag{1}$$

where $\theta$ denotes the parameters of the writer. It depends on the architecture of the decoder whether, given $z_t$, the prediction of $y_t$ further depends on the full decoding $\boldsymbol{z}_{<t}$ path that led up to this point.

### 3.1.1    CONVOLUTION-BASED LOW-LATENCY DECODER

In order for the decoder to be trained efficiently for different source/target context sizes, we build upon the "pervasive attention" architecture of Elbayad et al. (2018). Their machine translation model

uses masked 2D convolutions across a source-target grid, as in Figure 1. The masked convolutions ensure that only information from past target tokens is used to predict the next one. Here we adapt the masking pattern so as to construct the field-of-view in a way that it only extends over previous positions in both source and target dimensions. More concretely, we use a $11 \times 11$ filter where only the top-left $6 \times 6$ weights are non-zero. In this manner features computed by a sequence of convolutional layers at a position $(t, j)$ in the grid can be used to define $p_\theta(y_t | \boldsymbol{y}_{<t}, \boldsymbol{x}_{\leqslant j})$.

As input, at each site $(t, j)$, the network takes the concatenation of the (sub-)word embedding of the $t$-th token in target and the $j$-th token in the source sequence. The input is then progressively transformed along a number of convolutional network layers. Let $H = (h_{tj})$ for $1 \leqslant t \leqslant |\boldsymbol{y}|$ and $1 \leqslant j \leqslant |\boldsymbol{x}|$ be the final features of the masked convolutional neural network. Elbayad et al. (2018) apply max-pooling across the source dimension to map $H$ to a fixed-sized vector for each target position. To define $p_\theta(y_t | \boldsymbol{y}_{<t}, \boldsymbol{x}_{\leqslant j})$ in our low-latency model, we use

$$f_{tj} = \text{Pool}(H_{\leqslant t, \leqslant j}), \tag{2}$$

where $\text{Pool}(\cdot)$ pools $H_{\leqslant t, \leqslant j}$ in the source and/or target direction, or neither in which case $f_{tj} = h_{tj}$. Preliminary experiments with the pervasive attention architecture showed that max-pooling across the source dimension only is the best option. For the remainder of this paper we have:

$$f_{tj} = \text{Max-pool}(H_{t, \leqslant j}). \tag{3}$$

From there we generate the emission probabilities by linearly projecting the feature $f_{tj}$ to the dimension of the output vocabulary with the matrix $W$, followed by a soft-max normalization:

$$p_\theta(y_t | \boldsymbol{y}_{<t}, \boldsymbol{x}_{\leqslant j}) = \text{softmax}(W f_{tj}). \tag{4}$$

Due to the 2D convolutional structure of the network, the path leading up to position $(t, j)$ in the network does not impact the prediction of $y_t$. Therefore, we drop the dependence on $\boldsymbol{z}_{<t}$ in Eq. (4).

### 3.1.2 ATTENTION-BASED LOW-LATENCY DECODER

The second online decoder we consider is based on the "transformer" model of Vaswani et al. (2017). Both the encoder and decoder consist of a stack of blocks. Each block consists of a multi-head self-attention followed by a position-wise fully connected feed-forward network, or equivalently a $1 \times 1$ convolution. The decoder uses an additional multi-head attention block that ranges over the source token encodings.

Given a query vector $q_t$, an attention component aggregates a set of value vectors $v_j$ in a weighted sum, based on scoring a corresponding set of key vectors $k_j$ against the query. Using the dot-product as the score, we obtain the attention aggregate $a_t$ as

$$a_t = \sum_j \alpha_{tj} v_j, \qquad \alpha_{tj} = \frac{\exp e_{tj}}{\sum_j \exp e_{tj}}, \qquad e_{tj} = q_t^\top k_j. \tag{5}$$

The self-attention block is designed such that only previous output tokens can be attended over, since future tokens are not available when generating targets, ensuring the decoder is autoregressive (AR).

In our work, the transformer model for online translation has the following properties:

1. A uni-directional encoder: we make the self-attention in the encoder autoregressive, so that it can encode all source prefixes in parallel during training, and encode them progressively during generation. This is achieved by substituting the mask used in Ma et al. (2019) (referred to as STACL in Eq. (6)) with the AR mask of Eq. (7).

2. A constrained encoder-decoder interaction: we mask the source attention in the decoder, so that when producing the $t$-th output token attention is limited to the $z_t$ source tokens read so far. This is guaranteed with the enc-dec mask of Eq. (7).

$$\forall t, \; e_{ij}^{\text{STACL}} = \begin{cases} q_i^\top k_j, & \text{If } i, j \leqslant z_t \\ -\infty, & \text{otherwise} \end{cases}, \tag{6}$$

$$e_{ij}^{\text{AR}} = \begin{cases} q_i^\top k_j, & \text{If } j \leqslant i \\ -\infty, & \text{otherwise} \end{cases}, \qquad e_{tj}^{\text{enc-dec}} = \begin{cases} q_t^\top k_j, & \text{If } j \leqslant z_t \\ -\infty, & \text{otherwise} \end{cases}. \tag{7}$$

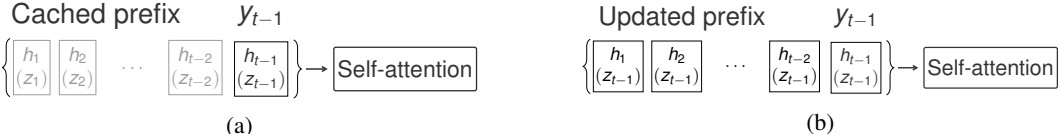

Figure 2: Caching (a) and updating (b) hidden states of the decoder for self-attention.

In this transformer-based decoder, the prediction of $y_t$ depends on the full decoding path $\boldsymbol{z}_{<t}$. This is because the decoder self-attention ranges over decoder hidden states of previous time steps $t' < t$, which themselves used $z_{t'}$ to attend over the source encoding.

In addition to the transformer-decoder described above, we consider a second transformer-based variant which removes the dependency of $y_t$ on $\boldsymbol{z}_{<t}$. To achieve this, each time a source token is read, we update all hidden states in the decoder based on the source context $\boldsymbol{x}_{z_t}$ available at that point. Essentially, we will re-run the decoder across the entire output sequence, using the new source context. This makes the decoder self-attention insensitive to $\boldsymbol{z}_{<t}$ given $z_t$, increasing the decoding cost from $O(|\boldsymbol{x}| \times |\boldsymbol{y}|)$ to $O(|\boldsymbol{x}|^2 \times |\boldsymbol{y}|)$. Note that the encoder still needs to be run only once, since its self-attention has been constrained to be autoregressive. We refer to this decoder as "transformer-update", see Figure 2 for an illustration.

### 3.2 TRAINING LOW-LATENCY DECODERS

For our decoders we use pre-defined "wait-$k$" schedules (Dalvi et al., 2018; Ma et al., 2019). First $k$ source tokens are read, before alternating reading and writing a single token at a time, until either the full source has been read, or the target generation has been terminated.

To train our models, we consider several strategies. The first, similar to Ma et al. (2019) is to train the model using the same wait-$k$ decoding path that will be used for generation. That is, to train the model, we optimize a sum of loss terms, each of which corresponds to the negative log-likelihood of predicting a target token:

$$L(\theta, \boldsymbol{x}, \boldsymbol{y}, \boldsymbol{z}) = -\sum_{t=1}^{|\boldsymbol{y}|} \ln p_\theta(y_t | \boldsymbol{y}_{<t}, \boldsymbol{x}_{\leqslant z_t}, \boldsymbol{z}_{<t}). \tag{8}$$

Training with $k=\infty$ corresponds to an offline (wait-until-the-end) decoder, with the exception that here we still use autoregressive dependencies in the encoder or along the source dimension in the 2D convolutional network.

In addition to training according to a single wait-$k$ decoding path, we can use losses associated with multiple paths. The additional loss terms may provide a richer training signal, and potentially yield models that could perform well across a range of values for $k$ during decoding. Due to the dependence of $y_t$ on the full decoding path $\boldsymbol{z}_{<t}$ in the transformer-based model, it is not possible to improve over simply training in parallel across the different values of $k$. When training for multiple values of $k$, we encode the source sentence once, and then forward it multiple times in the decoder, once for each value of $k$.

The 2D CNN-based architecture, however, does not include this full dependency, and allows parallel computation of all $p_\theta(y_t | \boldsymbol{y}_{<t}, \boldsymbol{x}_{\leqslant j})$ in single evaluation of the CNN. For this architecture we therefore use a loss of the form

$$L(\theta, \boldsymbol{x}, \boldsymbol{y}, q) = -\sum_{t=1}^{|\boldsymbol{y}|} \sum_{j=1}^{|\boldsymbol{x}|} q_{tj} \ln p_\theta(y_t | \boldsymbol{y}_{<t}, \boldsymbol{x}_{\leqslant j}), \tag{9}$$

where $q_{tj}$ weight the terms corresponding to different positions in the decoding grid of Figure 1. The weights $q_{tj}$ can be set to zero/one to reflect all positions corresponding to wait-$k$ decoding paths for one or more values of $k$. In our experiments we also consider the option of activating all loss terms above a certain wait-$k$ decoding path. The latter corresponds to the sum of all loss terms on all wait-$k'$ decoding paths with $k \leqslant k' \leqslant |\boldsymbol{x}|$.[1]

---

[1]Summing the losses for all paths, though, would weight shared loss terms more importantly.

The transformer-update architecture also avoids the full dependency on $\boldsymbol{z}_{<t}$, but in order to train from all loss terms above a certain decoding path, it requires a separate decoder run for each source context size, *i.e.* for each column of the decoding grid in Figure 1.

## 4 EXPERIMENTAL EVALUATION

In this section we present our experimental setup, followed by quantitative results.

### 4.1 DATASETS AND EXPERIMENTAL SETUP

**Datasets.** We evaluate our approach on IWSLT14 En-De (Cettolo et al., 2014) and on the WMT15 De-En datasets.[2] Replicating the setup of Elbayad et al. (2018) for IWSLT14 De-En, we train on 160K sentence pairs, develop on 7K held-out pairs and test on 6.7K pairs. The vocabulary consists of 8.8K types on the source side and 6.6K types on the target side that were obtained from a joint source and target byte pair encoding (BPE; Sennrich et al. 2016). For WMT15 De-En we use a joint vocabulary of 32k BPE types. We train on 4.5M pairs, develop on newstest2013 (3K pairs) and test on newstest15 (2.2K pairs).

**Evaluation metrics.** We use beam-search decoding, with a beam of size 5, for offline models and only greedy decoding online. We evaluate the translation quality by measuring case-sensitive tokenized BLEU (Papineni et al., 2002) with `multi-bleu.pl`.[3] Cho & Esipova (2016) measure the decoding latency using the proportion of the source sentence which has been read when producing target tokens, and average this proportion across the tokens in the generated sentence:

$$\text{AP} = \frac{1}{|\boldsymbol{y}|} \sum_{t}^{|\boldsymbol{y}|} z_t/|\boldsymbol{x}|. \tag{10}$$

More recently, average lagging (AL) and differentiable average lagging (DAL), were proposed to measure translation latency (Ma et al., 2019; Cherry & Foster, 2019; Arivazhagan et al., 2019):

$$\text{AL} = \frac{1}{\tau} \sum_{t}^{\tau} z_t - \frac{t-1}{\gamma}, \qquad \text{DAL} = \frac{1}{|\boldsymbol{y}|} \sum_{t}^{|\boldsymbol{y}|} z_t' - \frac{t-1}{\gamma}, \qquad z_t' = \begin{cases} z_t \text{ if } t = 1 \\ \max(z_t, z_{t-1}' + \frac{1}{\gamma}) \end{cases},$$

where $\gamma = |\boldsymbol{y}|/|\boldsymbol{x}|$ and $\tau = \min\{t \,|\, g(t) = |\boldsymbol{x}|\}$. These metrics handle differing source and target lengths more properly, and have an intuitive interpretation as the average by which the system lags behind an ideal instantaneous translator. We refer the reader to (Cherry & Foster, 2019) for details.

We report the mean of these metric across translations. In this section we report the average lagging (AL) metric, while the other metrics (AP, DAL, leading to a similar trend) are given in Appendix B. AL is chosen to make our results comparable with those of Ma et al. (2019) and Zheng et al. (2019b) for German-English WMT15 dataset.

**Architectures.** For the pervasive attention model, to reduce the memory footprint, we use residual skip connections (He et al., 2016b) rather than the dense connections (Huang et al., 2017) used by Elbayad et al. (2018). See Appendix A for more details about other minor changes made to the original architecture. We consider an offline pervasive attention baseline (PA) that reads the source sequence bi-directionally via asymmetric convolution filters that are only masking the future targets. Note that this baseline is not directly applicable for low-latency decoding, due to the bidirectional source encoding. We use a second baseline where convolutions are masked along the source dimension, which we refer to as masked-PA baseline (MPA).

For the Transformer model, we use a *small* architecture on IWSLT with an embedding dimension $d_{\text{enc}} = 512$ for the encoder, $d_{\text{dec}} = 256$ for the decoder, and $N = 6, d_{\text{ff}} = 1024, h = 4, P_{\text{drop}} = 0.3$. On WMT, we use a Transformer *base* (Vaswani et al., 2017) with tied embeddings. Similar to the pervasive attention baseline, we consider the offline baseline (T) with bidirectional self-attention in the source side and a masked baseline (MT) with left-to-right self-attention in the encoder.

---

[2] http://www.statmt.org/wmt15/
[3] https://github.com/moses-smt/mosesdecoder/blob/master/scripts/generic/multi-bleu.perl

| Datatset | task | PA* | PA | | MPA | | T | | MT | |
| | | BS | G | BS | G | BS | G | BS | G | BS |
|---|---|---|---|---|---|---|---|---|---|---|
| IWSLT14 | En-De | 27.21 | 27.23 | 27.96 | 26.66 | 27.52 | 27.46 | **28.27** | 26.58 | 27.70 |
| IWSLT14 | De-En | 33.86 | 33.06 | 34.02 | 32.21 | 33.39 | 33.84 | **34.94** | 32.81 | 33.79 |
| WMT15 | De-En | - | - | - | 28.08 | 29.28 | 31.96 | **33.00** | 31.14 | 32.27 |

Table 1: Offline evaluation of the Pervasive Attention (PA), Masked Pervasive Attention (MPA), Transformer (T) and masked Transformer (MT) baselines. PA* corresponds to results reported by Elbayad et al. (2018). We evaluate using greedy decoding (G) and beam-search (BS).

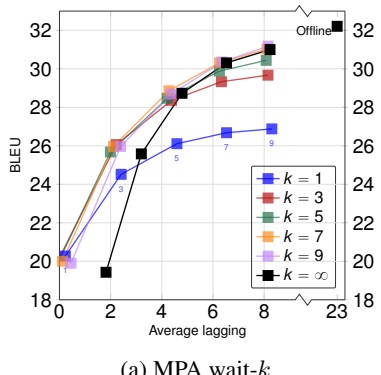
(a) MPA wait-$k$

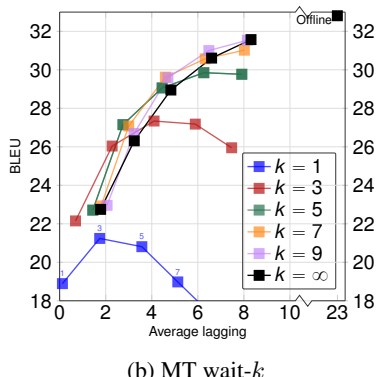
(b) MT wait-$k$

Figure 3: IWSLT De-En: wait-$k$ online decoding with MPA (a) and MT (b). Each curve is obtained with a model trained on a single decoding path. Each model is evaluated with $k_{\text{eval}} \in \{1, 3, 5, 7, 9\}$.

## 4.2 OFFLINE DECODING BASELINE RESULTS

In Table 1 we report the offline performance of the Pervasive Attention (PA), Masked Pervasive Attention (MPA), Transformer (T) and Masked Transformer (MT) baselines. Each with greedy decoding (G) and with beam search (BS).

Interestingly, the masked and non-masked versions perform quite similar, across both datasets and architectures. This validates the use of autoregressive source encoders in the low-latency models we consider below. The transformer and convolutional architectures perform similarly on IWSLT14, while the former performs significantly better on WMT15.

## 4.3 ONLINE DECODING RESULTS

In our first experiment we evaluate the wait-$k$ online decoding for different MT and MPA models, trained for different wait-$k$ decoding paths. We denote with $k=\infty$ the wait-until-end training where the full source is read before decoding. In each figure the offline results are added for reference, the offline model has a latency of $AL = |\boldsymbol{x}|$. Here we show results for IWSLT De-En, similar trends are observed for IWSLT En-De (see Appendix B.3).

**Impact of the architecture.** Figure 3 presents the performance of models trained for wait-$k$ decoding across a range of latencies $k_{\text{eval}} \in \{1, 3, 5, 7, 9\}$. Each trained model is represented by a curve, by evaluating it across different wait-$k$ decoding paths.

For the convolutional architecture we observe that a single model trained for a relatively large value of $k$, $e.g.$ 7 or 9, provides good performance when using it to generate using different $k$ values for generation. The performance of the transformer-based model is more sensitive to the value of $k$ used for training. This difference in sensitivity could be due to the use of input-independent convolution filters $vs.$ self-attention which relies on scoring context to weight its features.

Given an appropriate value for $k$ during training, the transformer-based models yield better translations for a given latency level.

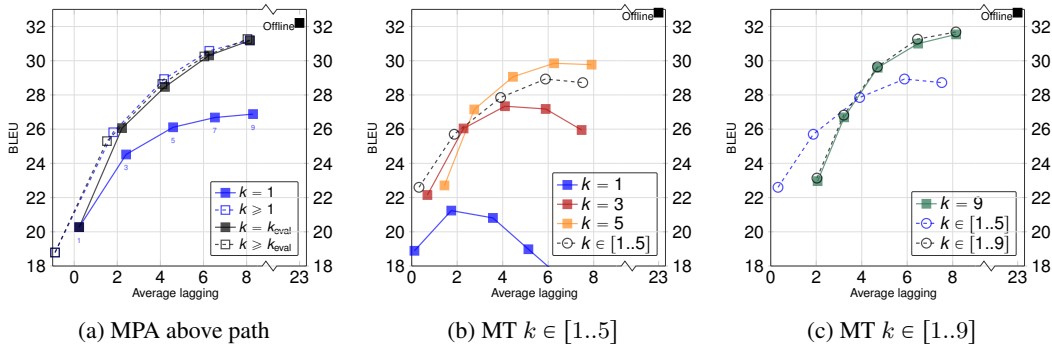

(a) MPA above path      (b) MT $k \in [1..5]$      (c) MT $k \in [1..9]$

Figure 4: IWSLT De-En: wait-$k$ online decoding with MT trained on a single path or multiple paths. Each model is evaluated with $k_{\text{eval}} \in \{1, 3, 5, 7, 9\}$. The $k = k_{\text{eval}}$ curve combines different models, evaluating for each value of $k$ the model trained for that value.

| Training | Update | $k_{\text{eval}} = 1$ | $k_{\text{eval}} = 3$ | $k_{\text{eval}} = 5$ | $k_{\text{eval}} = 7$ | $k_{\text{eval}} = 9$ |
|---|---|---|---|---|---|---|
| $k = \infty$ | w/o | 22.8 | 26.3 | 29.0 | 30.6 | 31.6 |
| $k = \infty$ | with | **24.7** | **27.8** | **29.9** | **31.4** | **32.1** |
| $k = 7$ | w/o | 22.9 | 27.1 | 29.6 | 30.6 | **31.0** |
| $k = 7$ | with | **24.7** | **27.9** | **30.0** | **31.1** | 31.0 |
| $k \in [1..5]$ | w/o | 22.6 | 25.7 | 27.9 | 28.9 | **28.7** |
| $k \in [1..5]$ | with | **25.0** | **28.0** | **29.1** | **29.1** | 28.6 |
| $k \in [1..9]$ | w/o | 23.1 | 26.8 | 29.6 | 31.3 | **31.7** |
| $k \in [1..9]$ | with | **24.5** | **28.0** | **30.1** | **31.4** | 31.7 |

Table 2: IWSLT De-En: Impact of hidden state updates in transformer-based low-latency decoders.

**Impact of multiple path training.** For the convolutional model (MPA), we consider activating loss terms in the area above the wait-$k$ path. Training on this area covers a combinatorially large number of paths with latency higher than $k$. The results in Figure 4a show that for the MPA architecture, multiple path training can be beneficial compared to single path ($k=k_{\text{eval}}$ vs. $k \geqslant k_{\text{eval}}$). More importantly, the model trained with all loss terms above the diagonal of the decoding grid ($k \geqslant 1$), performs equal or better than models trained with only a subset of the terms, across all tested latency levels.

For transformer model (MT), training for all decoding paths above a certain wait-$k$ path is prohibitively costly (since the cost is linear in the number of training paths). Therefore, we opted instead for joint training for a selected set of wait-$k$ paths. Results in Figure 4b show that jointly training on the wait-$k$ paths from $k=1$ to $k=5$ improves over the training on individual paths in low-latency regimes ($k_{\text{eval}} \leqslant 3$). For higher latency a relatively small drop in performance is observed compared to training with $k = 5$.

Where the convolutional model could be trained near optimal by training across all values of $k$, in Figure 4c we see that training the paths from $k=1$ to $k=9$ performs similarly to just training with $k=9$. The latter model is (near) optimal for AL$\geqslant 3$, but for very low-latency the model trained on $k \in [1..5]$ is better.

**Impact of hidden state updates.** The standard decoding approach with a Transformer is to use previously evaluated hidden states to encode the target prefix written so far. This is not the case for MPA where new source context is integrated in the convolution for all previous time-steps. To bridge the gap between the two decoding paradigms, we update previous hidden states to account for new source contexts.

In Table 2 we compare transformer models with and without hidden state updates, trained along one or multiple decoding paths. Among the 20 comparisons (4 models, and 5 wait-$k$ decoders) we

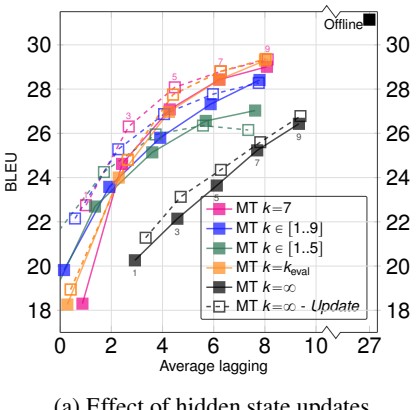
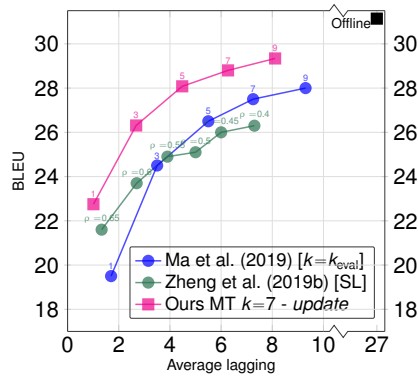

(a) Effect of hidden state updates

(b) Comparison to the state of the art

Figure 5: Experimental results on WMT15 De-En.

observe only a single case where state updates drop the BLEU score by 0.1. In all other cases, the state updates lead to improvements, by up to 2.4 BLEU points in very low latency settings.[4]

**Experiments on the WMT15 De-En corpus.** In Figure 5a we present our results on the WMT15 De-En task using transformer model, which gave the best results in the IWSLT experiments. The results confirm observations made on IWSLT: (i) Hidden state updates consistently improve the performance of the tested models. (ii) When using hidden state updates, a single low-latency model ($k = 7$) performs better or comparable to using separate models trained specifically for the value of $k$ used for decoding ($k_{eval}$).

The main differences w.r.t. the IWSLT experimental results are observed for the model trained for offline decoding ($k = \infty$). Whereas on IWSLT it was competitive with the best models trained for online decoding, for this larger corpus with longer sentences, this is no longer the case and the offline model performs significantly worse than the ones specifically trained for online decoding.

**Comparison to state-of-the-art.** Figure 5b compares our results with state-of-the-art performance reported by Ma et al. (2019) and Zheng et al. (2019b) for German-English translation on the WMT15 dataset. Our model ($k = 7$, with update) establishes a new state of the art for this task, significantly improving over previous work across the full range of latency levels.

It is important to note that to obtain our results we use a single trained model regardless of the latency level considered at decoding time, like Zheng et al. (2019b) but unlike Ma et al. (2019). Moreover, we use simple wait-$k$ decoders to attain different latency levels. Zheng et al. (2019b), on the other hand, trained a data adaptive controller to decide on the read/write actions, but used an underlying decoder trained for offline translation. Given our results, we expect that further gains are possible by training controllers on top of our model, which is trained specifically for online decoding.

**Training time.** We compare in Table 3 the training time of our models to the training time of the baseline and to that of our implementation of the original wait-k paper (Ma et al., 2019). If training on a wait-$k$ path, we report the average training time of $k \in \{1, 3, 5, 7, 9\}$. Updating either the encoder states to use bi-directionally or the prefix decoder states increases the training time dramatically. In fact, when training along the wait-$k$ path, the encoder update scales the encoder forward-passing time by $\min(|\boldsymbol{x}|, |\boldsymbol{y}|) - k + 1$ to produce states for each context size. Similarly, the decoder update increases the decoder forward-passing time by the same factor to re-compute the states with growing context.

**Decoding speed.** We compare in Table 4 the decoding speed of our models with and without updating the decoder states. We also include in the comparison the decoding speed if we update the encoder states as we advance in reading the source sequence, similar to Ma et al. (2019). To factor out the training approach, we only consider the models trained for offline decoding (MT and T of

---

[4]Table 2 does not report the AL values, which are comparable across different models given a fixed wait-$k$ decoder, *i.e.* per column. See Appendix B for results with AL values.

| Dataset | Model | Training time |
|---|---|---|
| | Our Wait-$k$ | 2.9 hours |
| | Our wait-$k$ with decoder update in training | 121 hours |
| IWSLT'14 De-En | Wait-$k$ with encoder update in training (Ma et al., 2019) | 116 hours |
| | Uni-directional baseline (MT) | 3.5 hours |
| | Bi-directional baseline (T) | 3.4 hours |
| | Our Wait-$k$ | 389 hours |
| WMT'15 De-En | Uni-directional baseline (MT) | 559 hours |
| | Bi-directional baseline (T) | 613 hours |

Table 3: Training time of our models compared to the baseline: IWSLT'14 De-En models were trained on a single GPU with a batch-size of 4k tokens ($\times 2$ gradient accumulation), and WMT'15 De-En models were trained on 2 GPUs with a batch-size of 3.5k tokens ($\times 64$ gradient accumulation).

Table 1) and evaluate them on the wait-$k$ paths. The reported speeds are the average of evaluating with $k \in \{1, 3, 5, 7, 9\}$.

| Dataset | Model | Update | GPU | CPU |
|---|---|---|---|---|
| IWSLT'14 De-En | MT | $\varnothing$ | 21.7k tokens/s | 130.3 tokens/s |
| | | decoder | 13.4k tokens/s | 97.3 tokens/s |
| | T | encoder | 7.3k tokens/s | 54.2 tokens/s |
| | | encoder+decoder | 5.5k tokens/s | 46.3 tokens/s |
| WMT'15 De-En | MT | $\varnothing$ | 6.3k tokens/s | 77.1 tokens/s |
| | | decoder | 3.5k tokens/s | 40.7 tokens/s |
| | T | encoder | 2.9k tokens/s | 31.9 tokens/s |
| | | encoder+decoder | 1.6k tokens/s | 20.4 tokens/s |

Table 4: Decoding speed in tokens/s on both GPU (RTX2080Ti) and CPU. For each model we measure the decoding speed with and without updating the decoder states.

The decoding speeds show that updating the prefix decoder states costs less than updating the encoder states with a bi-directional encoder on both GPU (IWSLT: $\times 1.8$, WMT: $\times 1.2$ speed-up) and CPU (IWSLT: $\times 1.8$ , WMT: $\times 1.3$ speed-up).

## 5 CONCLUSION

We compared transformer and 2D convolutional architectures for online machine translation with "wait-$k$" decoders, and proposed improved training techniques for these models. We find the transformer architecture to perform best, and improved its performance using hidden state updates inspired from the 2D convolutional architecture. We find that training a single model for relatively high values of $k$, *e.g.* 7 or 9, yields a performance that is comparable to using separate models trained for each specific value of $k$. Training a single model for multiple values of $k$ improves performance for very low-latency regimes.

Our results establish a new state of the art for online machine translation on WMT15 De-En dataset. We improve over recent results obtained with "wait-$k$" decoders (Ma et al., 2019) and trained data-adaptive controllers (Zheng et al., 2019b) to schedule read/write actions. Our results, together with those of Zheng et al. (2019b), suggest that further performance improvements are possible by integrating our models trained for online decoding with data-adaptive controllers.

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

## A  DETAILS ON THE CONVOLUTIONAL ARCHITECTURE

In this section we provide more details on the modifications we made to the convolutional "pervasive attention" architecture of Elbayad et al. (2018). In Figure 6 we provide a schematic overview of the original convolutional architecture of Elbayad et al. (2018).

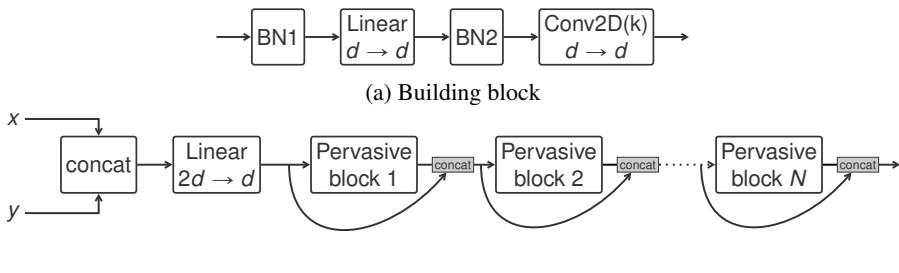

(a) Building block

(b) Overall dense architecture

Figure 6: Original convolutional architecture of Elbayad et al. (2018).

In our work, we made the following modifications to reduce the memory footprint of the model and to improve its performance.

1. We added feed-forward $1 \times 1$ convolutional layers after the masked convolutions for their important role in existing encoder architectures as they boost the representational power of the model.
2. We substituted the dense layer connections with residual ones to reduce the size of the features that keep increasing and costing more memory.
3. We opted for layer-normalization (Ba et al., 2016) instead of batch-normalization as it is more stable and more appropriate for causal sequences.
4. We use depth-wise separable convolutions for Conv2D(k) instead of ordinary convolutions.

In Figure 7 we provide a schematic overview of the adapted convolutional architecture.

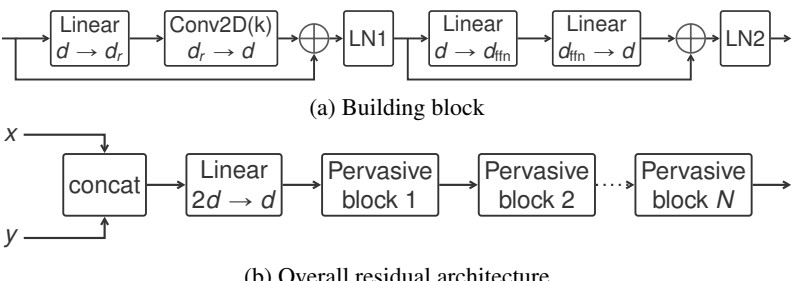

(a) Building block

(b) Overall residual architecture

Figure 7: Overview of the adapted convolutional architecture.

Finally, we found that removing the layer normalizations all-together and summing the block outputs before projecting on the target vocabulary to give best results. The final high-level architecture with block output addition is given in Figure 8.

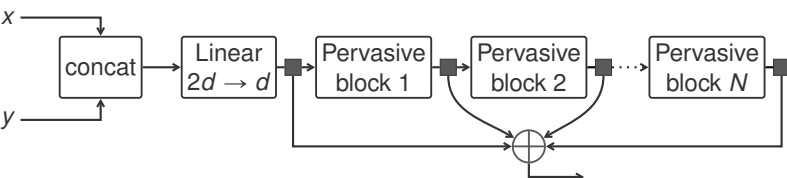

Figure 8: Overview of the architecture with block output addition used in this paper.

# B    ADDITIONAL EVALUATION RESULTS

In this appendix we provide additional evaluation results.

1. In addition to results in the main paper measuring latency with AL, in Appendix B.1 we provide results measuring latency using the AP and DAL metrics.

2. In Appendix B.2 we provide the numerical underlying the plots for IWSLT De-En in the main paper and appendix.

3. Finally, we provide results for the reverse En-De translation direction for both IWSLT in Appendix B.3 and WMT in Appendix B.4.

B.1   IWSLT DE-EN: AP AND DAL LATENCY

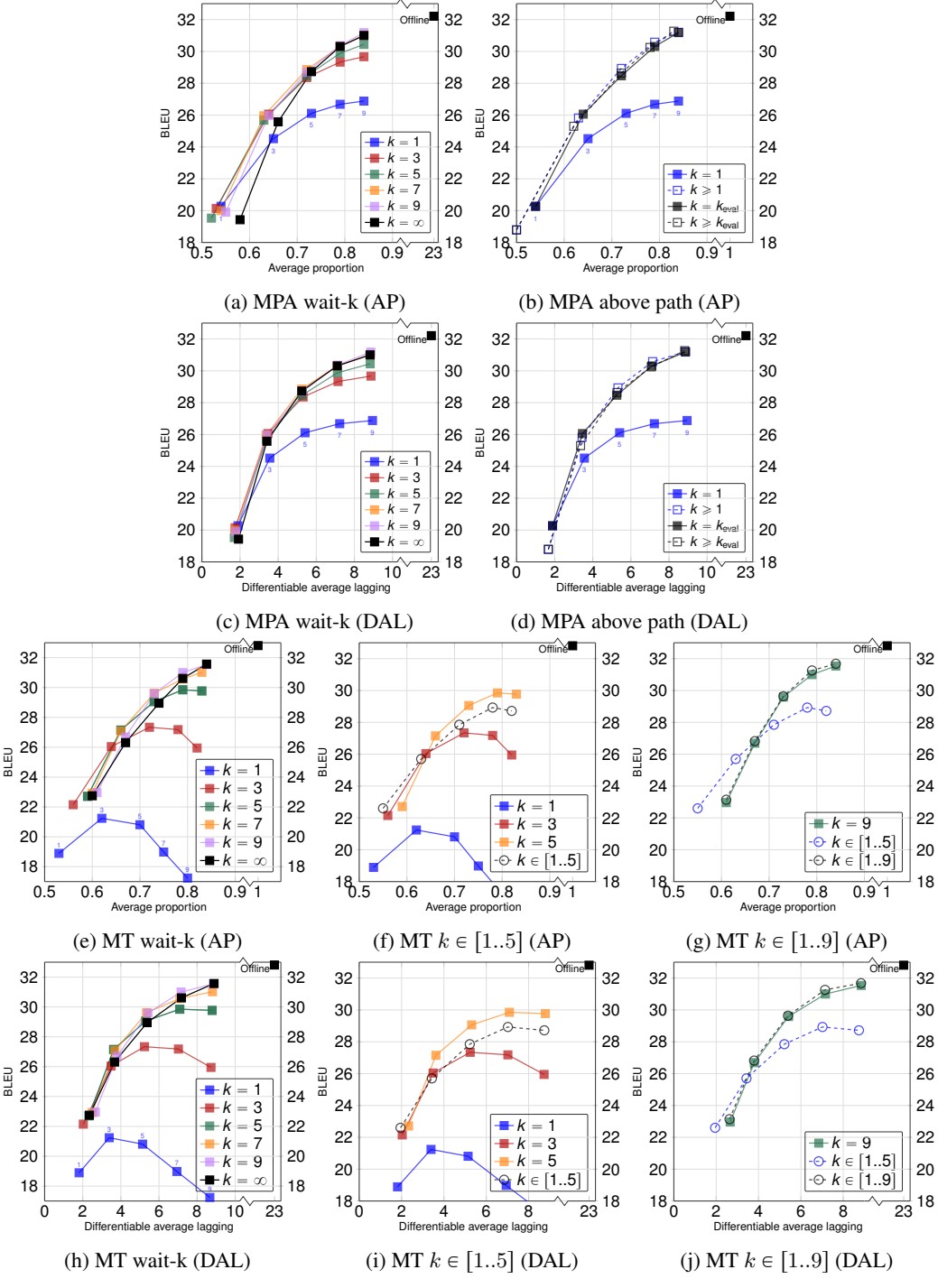

Figure 9: IWSLT De-En: Online decoding with single and multiple paths training. Measuring latency with AP and DAL, corresponding to figures 3 and 4 in the main paper where AL is used.

## B.2 IWSLT DE-E: NUMERICAL RESULTS

| k | BLEU | AP | AL | DAL |
|---|------|------|------|------|
| 1 | 18.9 | 0.53 | 0.1 | 1.8 |
| 3 | 21.2 | 0.62 | 1.7 | 3.4 |
| 5 | 20.8 | 0.70 | 3.6 | 5.1 |
| 7 | 19.0 | 0.75 | 5.1 | 6.9 |
| 9 | 17.2 | 0.80 | 6.6 | 8.7 |

(a) MT, $k = 1$

| k | BLEU | AP | AL | DAL |
|---|------|------|------|------|
| 1 | 22.8 | 0.60 | 1.8 | 2.4 |
| 3 | 26.3 | 0.67 | 3.2 | 3.7 |
| 5 | 29.0 | 0.74 | 4.8 | 5.4 |
| 7 | 30.6 | 0.79 | 6.6 | 7.2 |
| 9 | 31.6 | 0.84 | 8.3 | 8.9 |

(b) MT, $k = \infty$

| k | BLEU | AP | AL | DAL |
|---|------|------|------|------|
| 1 | 22.9 | 0.60 | 1.8 | 2.5 |
| 3 | 27.1 | 0.66 | 3.0 | 3.7 |
| 5 | 29.6 | 0.73 | 4.6 | 5.4 |
| 7 | 30.6 | 0.79 | 6.3 | 7.1 |
| 9 | 31.0 | 0.83 | 8.0 | 8.8 |

(c) MT, $k = 7$

| k | BLEU | AP | AL | DAL |
|---|------|------|------|------|
| 1 | 18.9 | 0.53 | 0.1 | 1.8 |
| 3 | 26.0 | 0.64 | 2.3 | 3.5 |
| 5 | 29.1 | 0.73 | 4.5 | 5.3 |
| 7 | 30.6 | 0.79 | 6.3 | 7.1 |
| 9 | 31.5 | 0.84 | 8.2 | 8.9 |

(d) MT, $k = k_{\text{eval}}$

| k | BLEU | AP | AL | DAL |
|---|------|------|------|------|
| 1 | 21.4 | 0.54 | 0.6 | 1.7 |
| 3 | 21.5 | 0.62 | 2.2 | 3.3 |
| 5 | 20.0 | 0.70 | 3.7 | 5.1 |
| 7 | 18.1 | 0.75 | 5.2 | 6.9 |
| 9 | 16.3 | 0.80 | 6.6 | 8.7 |

(e) MT, $k = 1$ - *Update*

| k | BLEU | AP | AL | DAL |
|---|------|------|------|------|
| 1 | 24.7 | 0.59 | 1.9 | 2.1 |
| 3 | 27.8 | 0.67 | 3.4 | 3.6 |
| 5 | 29.9 | 0.74 | 5.0 | 5.4 |
| 7 | 31.4 | 0.80 | 6.7 | 7.2 |
| 9 | 32.1 | 0.84 | 8.4 | 8.9 |

(f) MT, $k = \infty$ - *Update*

| k | BLEU | AP | AL | DAL |
|---|------|------|------|------|
| 1 | 24.7 | 0.58 | 2.0 | 2.2 |
| 3 | 27.9 | 0.66 | 3.3 | 3.6 |
| 5 | 30.0 | 0.73 | 4.9 | 5.4 |
| 7 | 31.1 | 0.79 | 6.6 | 7.1 |
| 9 | 31.0 | 0.84 | 8.2 | 8.8 |

(g) MT, $k = 7$ - *Update*

| k | BLEU | AP | AL | DAL |
|---|------|------|------|------|
| 1 | 21.4 | 0.54 | 0.6 | 1.7 |
| 3 | 27.1 | 0.65 | 2.9 | 3.5 |
| 5 | 28.8 | 0.73 | 5.0 | 5.5 |
| 7 | 31.1 | 0.79 | 6.6 | 7.1 |
| 9 | 31.7 | 0.84 | 8.3 | 8.9 |

(h) MT, $k = k_{\text{eval}}$ - *Update*

| k | BLEU | AP | AL | DAL |
|---|------|------|------|------|
| 1 | 18.8 | 0.50 | $-0.9$ | 1.7 |
| 3 | 25.8 | 0.63 | 1.8 | 3.4 |
| 5 | 28.9 | 0.72 | 4.2 | 5.3 |
| 7 | 30.6 | 0.79 | 6.3 | 7.1 |
| 9 | 31.2 | 0.84 | 8.1 | 8.9 |

(i) MPA, $k \geqslant 1$

| k | BLEU | AP | AL | DAL |
|---|------|------|------|------|
| 1 | 17.5 | 0.49 | $-1.0$ | 1.6 |
| 3 | 25.2 | 0.62 | 1.7 | 3.4 |
| 5 | 28.6 | 0.72 | 4.1 | 5.3 |
| 7 | 30.5 | 0.79 | 6.2 | 7.1 |
| 9 | 31.3 | 0.84 | 8.1 | 8.8 |

(j) MPA, $k \geqslant 5$

| k | BLEU | AP | AL | DAL |
|---|------|------|------|------|
| 1 | 17.2 | 0.49 | $-1.1$ | 1.5 |
| 3 | 24.5 | 0.62 | 1.4 | 3.3 |
| 5 | 28.3 | 0.71 | 3.9 | 5.2 |
| 7 | 30.3 | 0.78 | 6.0 | 7.1 |
| 9 | 31.3 | 0.83 | 8.0 | 8.8 |

(k) MPA, $k \geqslant 7$

| k | BLEU | AP | AL | DAL |
|---|------|------|------|------|
| 1 | 22.6 | 0.55 | 0.3 | 2.0 |
| 3 | 25.7 | 0.63 | 1.9 | 3.4 |
| 5 | 27.9 | 0.71 | 3.9 | 5.2 |
| 7 | 28.9 | 0.78 | 5.9 | 7.0 |
| 9 | 28.7 | 0.82 | 7.5 | 8.7 |

(l) MT, $k \in [1..5]$

| k | BLEU | AP | AL | DAL |
|---|------|------|------|------|
| 1 | 23.1 | 0.60 | 1.8 | 2.5 |
| 3 | 27.2 | 0.66 | 3.0 | 3.7 |
| 5 | 29.7 | 0.73 | 4.5 | 5.4 |
| 7 | 30.7 | 0.79 | 6.3 | 7.1 |
| 9 | 31.1 | 0.83 | 8.1 | 8.8 |

(m) MT, $k \in [3..7]$

| k | BLEU | AP | AL | DAL |
|---|------|------|------|------|
| 1 | 23.1 | 0.61 | 2.0 | 2.6 |
| 3 | 26.8 | 0.67 | 3.2 | 3.8 |
| 5 | 29.6 | 0.73 | 4.7 | 5.4 |
| 7 | 31.3 | 0.79 | 6.5 | 7.1 |
| 9 | 31.7 | 0.84 | 8.1 | 8.8 |

(n) MT, $k \in [1..9]$ - *Update*

| k | BLEU | AP | AL | DAL |
|---|------|------|------|------|
| 1 | 25.0 | 0.56 | 1.3 | 1.9 |
| 3 | 28.0 | 0.65 | 2.8 | 3.5 |
| 5 | 29.1 | 0.72 | 4.6 | 5.3 |
| 7 | 29.1 | 0.78 | 6.2 | 7.0 |
| 9 | 28.6 | 0.83 | 7.8 | 8.8 |

(o) MT, $k \in [1..5]$ - *Update*

| k | BLEU | AP | AL | DAL |
|---|------|------|------|------|
| 1 | 23.1 | 0.60 | 1.8 | 2.5 |
| 3 | 27.2 | 0.66 | 3.0 | 3.7 |
| 5 | 29.7 | 0.73 | 4.5 | 5.4 |
| 7 | 30.7 | 0.79 | 6.3 | 7.1 |
| 9 | 31.1 | 0.83 | 8.1 | 8.8 |

(p) MT, $k \in [3..7]$ - *Update*

| k | BLEU | AP | AL | DAL |
|---|------|------|------|------|
| 1 | 24.5 | 0.59 | 2.1 | 2.3 |
| 3 | 28.0 | 0.67 | 3.4 | 3.7 |
| 5 | 30.1 | 0.74 | 5.0 | 5.4 |
| 7 | 31.4 | 0.79 | 6.7 | 7.2 |
| 9 | 31.7 | 0.84 | 8.3 | 8.9 |

(q) MT, $k \in [1..9]$ - *Update*

Table 5: Numerical results of IWSLT De-En

## B.3 IWSLT EN-DE

| k | BLEU | AP | AL | DAL |
|---|------|------|------|------|
| 1 | 18.5 | 0.61 | 2.5 | 2.8 |
| 3 | 22.8 | 0.69 | 3.8 | 4.1 |
| 5 | 25.0 | 0.75 | 5.5 | 5.7 |
| 7 | 25.9 | 0.81 | 7.2 | 7.5 |
| 9 | 26.2 | 0.86 | 8.9 | 9.2 |

(a) MT, $k = \infty$

| k | BLEU | AP | AL | DAL |
|---|------|------|------|------|
| 1 | 18.7 | 0.62 | 2.6 | 3.0 |
| 3 | 23.2 | 0.69 | 3.9 | 4.2 |
| 5 | 25.4 | 0.75 | 5.4 | 5.8 |
| 7 | 26.4 | 0.81 | 7.1 | 7.4 |
| 9 | 26.4 | 0.85 | 8.7 | 9.1 |

(b) MT, $k = 7$

| k | BLEU | AP | AL | DAL |
|---|------|------|------|------|
| 1 | 16.4 | 0.59 | 1.9 | 2.5 |
| 3 | 22.1 | 0.68 | 3.7 | 4.0 |
| 5 | 25.4 | 0.75 | 5.3 | 5.7 |
| 7 | 26.4 | 0.81 | 7.1 | 7.4 |
| 9 | 26.7 | 0.85 | 8.8 | 9.1 |

(c) MT, $k = k_{\text{eval}}$

| k | BLEU | AP | AL | DAL |
|---|------|------|------|------|
| 1 | 21.2 | 0.60 | 2.3 | 2.6 |
| 3 | 24.3 | 0.68 | 3.8 | 4.0 |
| 5 | 25.8 | 0.75 | 5.4 | 5.7 |
| 7 | 26.2 | 0.81 | 7.2 | 7.5 |
| 9 | 26.4 | 0.86 | 8.9 | 9.2 |

(d) MT, $k = \infty$ - *Update*

| k | BLEU | AP | AL | DAL |
|---|------|------|------|------|
| 1 | 21.4 | 0.60 | 2.3 | 2.5 |
| 3 | 24.7 | 0.68 | 3.7 | 3.9 |
| 5 | 26.2 | 0.75 | 5.3 | 5.6 |
| 7 | 26.3 | 0.81 | 7.0 | 7.3 |
| 9 | 26.0 | 0.85 | 8.6 | 9.0 |

(e) MT, $k = 7$ - *Update*

| k | BLEU | AP | AL | DAL |
|---|------|------|------|------|
| 1 | 16.6 | 0.57 | 1.6 | 2.1 |
| 3 | 22.9 | 0.67 | 3.5 | 3.7 |
| 5 | 25.6 | 0.75 | 5.2 | 5.5 |
| 7 | 26.3 | 0.81 | 7.0 | 7.3 |
| 9 | 26.2 | 0.85 | 8.6 | 9.0 |

(f) MT, $k = k_{\text{eval}}$ - *Update*

Table 6: Numerical results of IWSLT En-De

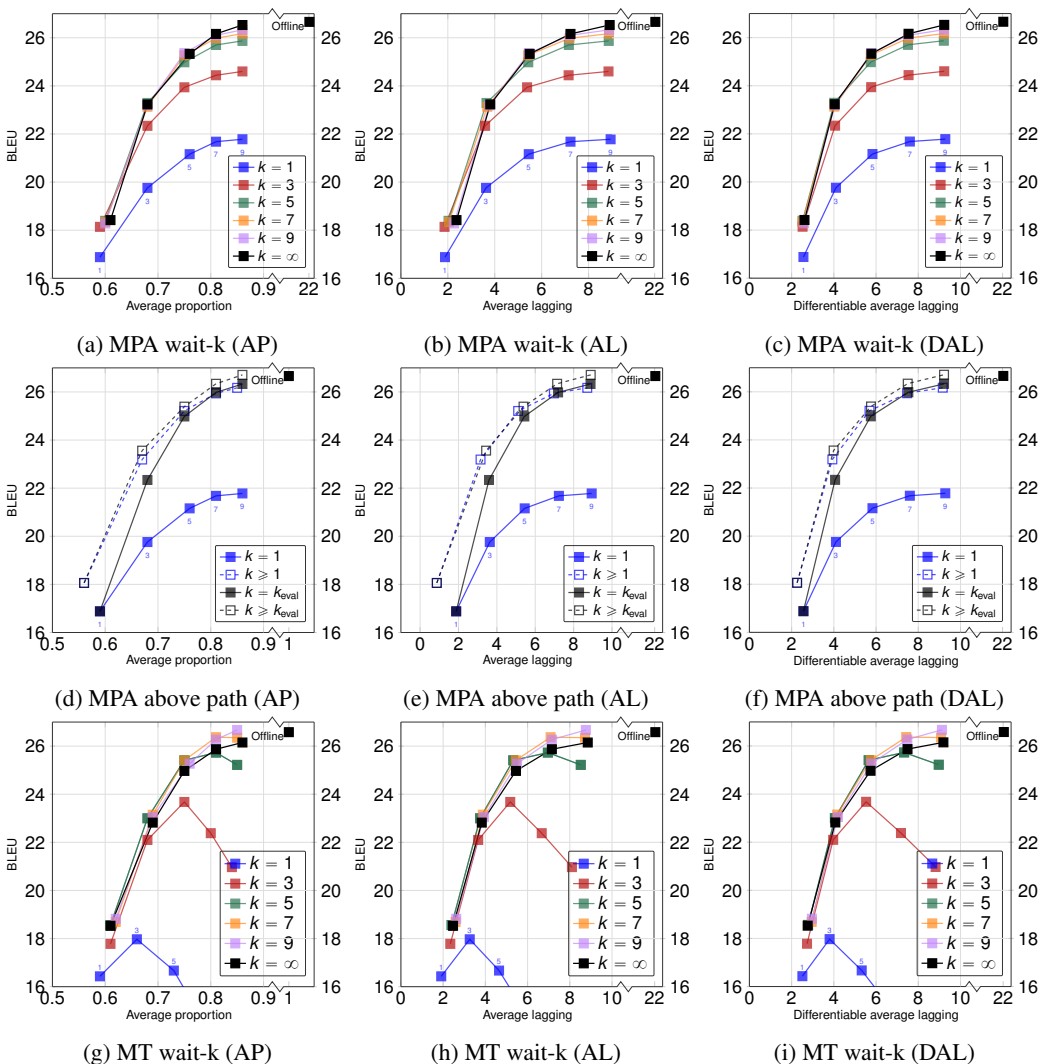

Figure 10: IWSLT En-De: Online decoding with single and multiple paths training. Measuring latency with AP, AL and DAL

## B.4 WMT EN-DE

| k | BLEU | AP | AL | DAL |
|---|------|-----|-----|-----|
| 1 | 18.3 | 0.53 | 0.3 | 2.1 |
| 3 | 24.0 | 0.61 | 2.3 | 3.6 |
| 5 | 27.0 | 0.68 | 4.3 | 5.4 |
| 7 | 28.4 | 0.75 | 6.2 | 7.2 |
| 9 | 29.3 | 0.79 | 8.0 | 9.0 |

(a) MT, $k = k_{\text{eval}}$

| k | BLEU | AP | AL | DAL |
|---|------|-----|-----|-----|
| 1 | 18.3 | 0.56 | 0.9 | 2.4 |
| 3 | 24.6 | 0.62 | 2.4 | 3.6 |
| 5 | 27.1 | 0.69 | 4.3 | 5.4 |
| 7 | 28.4 | 0.75 | 6.2 | 7.2 |
| 9 | 29.0 | 0.79 | 8.1 | 9.0 |

(b) MT, $k = 7$

| k | BLEU | AP | AL | DAL |
|---|------|-----|-----|-----|
| 1 | 21.3 | 0.62 | 3.3 | 4.1 |
| 3 | 23.1 | 0.68 | 4.7 | 5.4 |
| 5 | 24.4 | 0.73 | 6.3 | 7.0 |
| 7 | 25.6 | 0.78 | 7.8 | 8.5 |
| 9 | 26.8 | 0.82 | 9.4 | 10.0 |

(c) MT, $k = \infty$

| k | BLEU | AP | AL | DAL |
|---|------|-----|-----|-----|
| 1 | 18.9 | 0.53 | 0.4 | 2.0 |
| 3 | 24.8 | 0.62 | 2.6 | 3.6 |
| 5 | 27.8 | 0.69 | 4.4 | 5.4 |
| 7 | 28.8 | 0.75 | 6.3 | 7.2 |
| 9 | 29.4 | 0.79 | 8.0 | 9.0 |

(d) MT, $k = k_{\text{eval}}$ - *Update*

| k | BLEU | AP | AL | DAL |
|---|------|-----|-----|-----|
| 1 | 22.8 | 0.55 | 1.0 | 1.9 |
| 3 | 26.3 | 0.62 | 2.7 | 3.5 |
| 5 | 28.1 | 0.69 | 4.5 | 5.4 |
| 7 | 28.8 | 0.75 | 6.3 | 7.2 |
| 9 | 29.3 | 0.79 | 8.1 | 9.0 |

(e) MT, $k = 7$ - *Update*

| k | BLEU | AP | AL | DAL |
|---|------|-----|-----|-----|
| 1 | 21.3 | 0.62 | 3.3 | 4.1 |
| 3 | 23.1 | 0.68 | 4.7 | 5.4 |
| 5 | 24.4 | 0.73 | 6.3 | 7.0 |
| 7 | 25.6 | 0.78 | 7.8 | 8.5 |
| 9 | 26.8 | 0.82 | 9.4 | 10.0 |

(f) MT, $k = \infty$ - *Update*

Table 7: Numerical results of WMT De-En underlying figures 3 and 4 in the main paper.

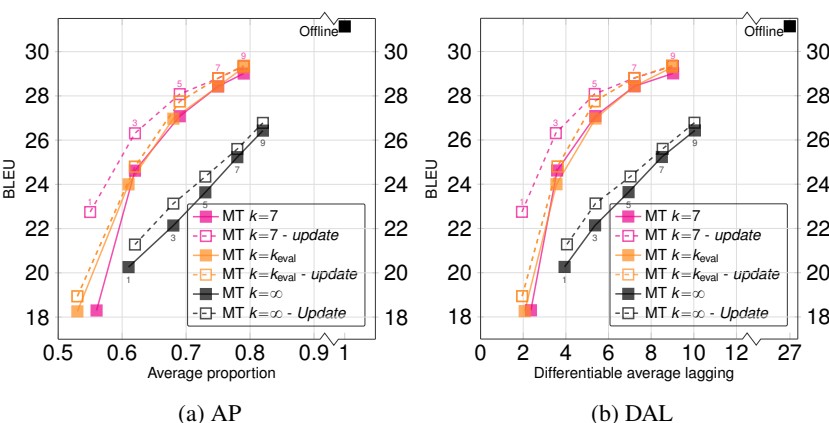

(a) AP

(b) DAL

Figure 11: WMT De-En results with latency measures AP and DAL.

