# OpenReview forum: "Improved Training Techniques for Online Neural Machine Translation"
_ICLR.cc/2020/Conference — Reject_

### Official Review · AnonReviewer2 · 2019-10-23
**Official Blind Review #2**

**Rating:** 3

**Review:**

This work apply the wait-k decoding policy on the 2D CNN-based architecture and transformer.  In the transformer-based model the author proposed to recalculate the decoder hidden states when a new source token arrives. The author also suggested to train with multiple k at the decoder level with shared encoder output. The experiments showed that the transformer model provide the best quality on IWSLT14 En-De, De-En, and WMT15 De-EN.

The masking and using causal attention for the transformer has been proposed in previous works. The hidden state updates provide some gains for the model but also makes the decoder more expensive. The training with multiple k provides similar gain as training with one k larger than the value used at the inference time. Overall the contributions are limited.

There is quite some room for this paper to improve its clarify, especially in terms of annotations and explaining the proposed ideas.

**Experience Assessment:**

I have published one or two papers in this area.

**Review Assessment: Checking Correctness Of Derivations And Theory:**

I carefully checked the derivations and theory.

**Review Assessment: Checking Correctness Of Experiments:**

I assessed the sensibility of the experiments.

**Review Assessment: Thoroughness In Paper Reading:**

I read the paper at least twice and used my best judgement in assessing the paper.

---

> ### Author Response · Authors · 2019-11-07
> **Thank you for the review and comments**
>
> Regarding "The masking and using causal attention for the transformer has been proposed in previous works."
>
> Uni-directional encoders for online machine translation were previously used with RNN-based architectures, we are not familiar with existing work that uses causal attention for transformer NMT models. In [1] the encoder was not causal which means that for every time-step the encoder states have to be updated. It would be great if you could share the references you were thinking about.
>
>
> Regarding "The hidden state updates provide some gains for the model but also makes the decoder more expensive."
>
> Compared to our model with caching, we agree that the update of the decoder states is expensive. However if we’re comparing to [1] we basically re-allocated the cost from updating the encoder states to updating the decoder states instead and we get better performances with this new allocation.
>
> Regarding "The training with multiple k provides similar gain as training with one k larger than the value used at the inference time. Overall the contributions are limited."
>
> Training with a single large k does not improve the performance on smaller values of k.
> If we look for example at figure 4.c, for an average lagging of 2, there is a difference of almost 3 BLEU points between the model trained with k=9 and the one trained with k in [1,...,5].
>
> Regarding "There is quite some room for this paper to improve its clarify, especially in terms of annotations and explaining the proposed ideas."
>
> Please let us know if there are any specific annotations or concepts you think need rewriting, we would gladly make it clearer in the updated paper.
>
> [1] Ma et al. “STACL: Simultaneous Translation with Implicit Anticipation and Controllable Latency using Prefix-to-Prefix Framework." ACL 2019

---

### Official Review · AnonReviewer1 · 2019-10-24
**Official Blind Review #1**

**Rating:** 3

**Review:**

This paper extends the idea of prefix-to-prefix in STACL and proposes two different variations. The authors did some interesting experiments between caching and updating decoder.

My questions are as follows:

1) in section 3.1.2, the authors mentioned two adaptations. Is the proposed AR encoder uni-directional? If the AR encoder is uni-directional, then I would be surprised that the uni-directional encoder outperforms the bi-directional encoder in the original wait-k model. For the second bullet, I think the original wait-k also did the same thing(they mentioned this in the paper clearly). So there is nothing new about bullet 2.

2) the idea mentioned in fig 1 is very similar to [1]. I suggest the authors compare with the aforementioned methods.

3) updating the hidden state of the decoder introduces more complexity during the inference time. I recommend the authors to perform some analysis about decoding time with CPU and GPU.

4) it is also interesting to show more comparison between different models' training time with the original STACL.

[1] Zheng et al. "Simultaneous Translation with Flexible Policy via Restricted Imitation Learning" ACL 2019

**Experience Assessment:**

I have published one or two papers in this area.

**Review Assessment: Checking Correctness Of Derivations And Theory:**

I assessed the sensibility of the derivations and theory.

**Review Assessment: Checking Correctness Of Experiments:**

I carefully checked the experiments.

**Review Assessment: Thoroughness In Paper Reading:**

I read the paper at least twice and used my best judgement in assessing the paper.

---

> ### Author Response · Authors · 2019-11-07
> **Thank you for the review and comments!**
>
> Regarding "1) in section 3.1.2, the authors mentioned two adaptations. Is the proposed AR encoder uni-directional? If the AR encoder is uni-directional, then I would be surprised that the uni-directional encoder outperforms the bi-directional encoder in the original wait-k model. "
>
> The original wait-k paper [2] and the subsequent paper [1,3] use an ‘incremental encoder’. It is bidirectional for all tokens before a cursor g(t). Therefore at every decoding time-step t, with increased g(t) the encoder states have to be updated. It is interesting, indeed perhaps surprising, to see that in the context of online machine translation the unidirectional encoder outperforms the bidirectional one. All of our transformer model for online translation use uni-directional encoders (MT) and when evaluated without the  ‘update’, the decoders are equivalent to the ones in [2] which suggests that the uni-directional encoders are better suited for online translation.
>
> Regarding "For the second bullet, I think the original wait-k also did the same thing(they mentioned this in the paper clearly). So there is nothing new about bullet 2. "
>
> Indeed, there is nothing new about the masking in the encoder-decoder interaction but we wanted to explicitly define all the masks used in the architecture, and list the differences with respect to an offline transformer model.
>
> Regarding "2) the idea mentioned in fig 1 is very similar to [1]. I suggest the authors compare with the aforementioned methods. "
>
> We will compare with the method in [1]. However In [1] the authors suggest optimizing the decoding along a set of paths sampled within an area of interest (similar to the gray area in our figure 2.) but ended up optimizing along the two boundary paths. What we suggest here is to optimize the decoding in ‘all’ the cells of the gray area. This is possible with the pervasive attention architecture where the cell state is independent from the path we followed to arrive there. With the transformer model, optimizing the full area is not evident, so we ended up selecting a few wait-k paths and thus using training strategy similar to the one in [1]. There is also the fact that [1] aims to learn dynamic read/write decision with a special token added to the vocabulary to represent the ‘Read’ action. Unfortunately, [1] only reports results for Chine-English (and reverse) translation, preventing direct comparison to  our results and those of [2,3].
>
> Regarding "3) updating the hidden state of the decoder introduces more complexity during the inference time. I recommend the authors to perform some analysis about decoding time with CPU and GPU."
>
> We will include decoding times on CPU and GPU for our models and compare them to the approach in [1].
>
> Regarding "4) it is also interesting to show more comparison between different models' training time with the original STACL. "
>
> During the training of our models on a given wait-k path we do not update the encoder states nor the previous decoder states. This makes our training time comparable to an offline transformer. With [1] however, given a target sequence y of length |y| there are |y| encoder forward passes to evaluate the states associated with each context size g(t).
>
> Between our single k training and multiple k, the training time is higher if for each sentence pair we run a separate forwards pass for each value of k. Alternatively, for each batch of sentence pairs we can sample a value of k and only use the loss for the wait-k path for that value of k for that batch. This way we end up with a comparable training times.
>
> [1] Zheng et al. "Simultaneous Translation with Flexible Policy via Restricted Imitation Learning" ACL 2019
> [2] Ma et al. “STACL: Simultaneous Translation with Implicit Anticipation and Controllable Latency using Prefix-to-Prefix Framework." ACL 2019
> [3] Zheng et al. “Simpler and faster learning of adaptive policies for simultaneous translation."  EMNLP 2019

---

### Official Review · AnonReviewer3 · 2019-10-27
**Official Blind Review #3**

**Rating:** 6

**Review:**

Sorry, this is a very quick review.

The paper is about an improved method of training latency-limited (wait-k) decoders for transformer-based machine translation, in which the right context is limited to various numbers.  So it's a kind of augmentation method that's well matched to the test scenario.  At least, that is my understanding.

I am not really an MT expert so cannot comment with much authority.  On the plus side the paper says it sets a new state of the art for latency-limited decoding for a German-English MT task, and it involves transformers, which are quite hot right now so the  attendees might find it interesting because of that connection.
On the minus side, it is all really quite task-specific.
I am putting weak accept.. regular-strength accept might be my other choice.
It's all with low confidence.

**Experience Assessment:**

I have read many papers in this area.

**Review Assessment: Checking Correctness Of Derivations And Theory:**

I did not assess the derivations or theory.

**Review Assessment: Checking Correctness Of Experiments:**

I did not assess the experiments.

**Review Assessment: Thoroughness In Paper Reading:**

I made a quick assessment of this paper.

---

### Author Response · Authors · 2019-11-14
**Paper updates**

Based on the suggestions of the reviewers, we made a few updates to the paper:
Made the comparison to the original wait-k paper [1] clearer, highlighting the differences in the encoder side.
Added training time details of our models as compared to the baselines and our implementation of STACL [1].
Added decoding speeds on GPU and CPU with and without decoder states update as well as the decoding speed of our implementation of [1].

[1] Ma et al. “STACL: Simultaneous Translation with Implicit Anticipation and Controllable Latency using Prefix-to-Prefix Framework." ACL 2019

---

### Decision · Program_Chairs · 2019-12-19

**Decision:**

Reject

**Comment:**

The paper proposes a method of training latency-limited (wait-k) decoders for online machine translation. The authors investigate the impact of the value of k, and of recalculating the transformer's decoder hidden states when a new source token arrives. They significantly improve over state-of-the-art results for German-English translation on the WMT15 dataset, however there is limited novelty wrt previous approaches. The authors responded in depth to reviews and updated the paper with improvements, for which there was no reviewer response. The paper presents interesting results but IMO the approach is not novel enough to justify acceptance at ICLR.